# Antioxidant and Anti-Inflammatory Effect of Cinnamon (*Cinnamomum verum* J. Presl) Bark Extract after In Vitro Digestion Simulation

**DOI:** 10.3390/foods12030452

**Published:** 2023-01-18

**Authors:** Stefania Pagliari, Matilde Forcella, Elena Lonati, Grazia Sacco, Francesco Romaniello, Pierangela Rovellini, Paola Fusi, Paola Palestini, Luca Campone, Massimo Labra, Alessandra Bulbarelli, Ilaria Bruni

**Affiliations:** 1ZooPlantLab, Department of Biotechnology and Biosciences, University of Milano-Bicocca, Piazza della Scienza 2, 20126 Milan, Italy; 2Department of Biotechnology and Biosciences, University of Milano-Bicocca, Piazza della Scienza 2, 20126 Milan, Italy; 3School of Medicine and Surgery, University of Milano-Bicocca, Via Cadore 48, 20900 Monza, Italy; 4Innovhub Stazioni Sperimentali per l’Industria S.r.l., Via Giuseppe Colombo 79, 20133 Milano, Italy

**Keywords:** cinnamon, antioxidant, oxidative stress, inflammation, in vitro digestion simulation, natural product, functional food

## Abstract

Cinnamon bark is widely used for its organoleptic features in the food context and growing evidence supports its beneficial effect on human health. The market offers an increasingly wide range of food products and supplements enriched with cinnamon extracts which are eliciting beneficial and health-promoting properties. Specifically, the extract of *Cinnamomum* spp. is rich in antioxidant, anti-inflammatory and anticancer biomolecules. These include widely reported cinnamic acid and some phenolic compounds, such asproanthocyanidins A and B, and kaempferol. These molecules are sensitive to physical-chemical properties (such as pH and temperature) and biological agents that act during gastric digestion, which could impair molecules’ bioactivity. Therefore, in this study, the cinnamon’s antioxidant and anti-inflammatory bioactivity after simulated digestion was evaluated by analyzing the chemical profile of the pure extract and digested one, as well as the cellular effect in vitro models, such as Caco2 and intestinal barrier. The results showed that the digestive process reduces the total content of polyphenols, especially tannins, while preserving other bioactive compounds such as cinnamic acid. At the functional level, the digested extract maintains an antioxidant and anti-inflammatory effect at the cellular level.

## 1. Introduction

Herbs and spices have been used since ancient times for nutritional and traditional medicine applications as they are rich in natural bioactive substances eliciting, for example, antioxidant and anti-inflammatory responses. Among these, cinnamon is widely used both as a bark extract and powder. There are over 300 species of cinnamon, but the most diffused as food/nutraceuticals are *Cinnamomum verum* J. Presl, *Cinnamomum aromaticum* or *Cinnamomum cassia* (L.) J. Presl, *Cinnamomum burmanni* (Nees and Th. Nees) Nees ex Blume, 2012 and *Cinnamomum loureiroi* Nees. These species differ in morphology and chemical composition. Specifically, at the phytochemical level, significant differences in the content of polyphenols and volatile phenols have been observed among various genotypes [1]. Moreover, the secondary metabolites profile also depends on plant growth conditions and the associated environmental pressures [2,3]. Among the most interesting compounds of bioactive Cinnamomum accessions, there are vanillic acid, caffeic acid, gallic acid, *p*-coumaric acid, ferulic acid, proanthocyanidins A and B, kaempferol, cinnamic acid and cinnamaldehyde, which exhibit several human beneficial effects, such as neuroprotective, hepatoprotective, cardioprotective and gastroprotective [1,4]. Most of these compounds are related to the antioxidant activity, enhancing the activities of catalase (CAT), superoxide dismutase (SOD) and glutathione peroxidase (GPx) [5,6]. Another important activity of cinnamon phyto-complexes is the anti-inflammatory one, which has been demonstrated in various cell and animal models and diseases, such as colitis, arthritis and diabetes [7,8,9,10].

Because of these healthful properties, the cinnamon extract and/or specific interesting compounds have been used as a bioactive ingredient in food products and supplements. However, it should be considered that the use of only isolated molecules is quite limited both because, in many cases, the beneficial effect is performed by a combination of molecules that act in synergy, and due to their poor systemic distribution and relative bioavailability [11]. For this reason, in recent years, the use of total cinnamon phytoextracts obtained by aqueous or hydroalcoholic extractions capable of optimizing the content of various bioactive molecules has become increasingly widespread. The extracts are often used to produce functional foods, such as yoghurt, creams and ice creams, rather than confectionery products up to food supplements and dietetic foods [12].

Starting from these premises, the present work was aimed at the study of the functional bioactivity of cinnamon extracts used in food supplements. Specifically, this paper analyzes the cinnamon bark, normally used in cooking, pulverized, to test the permanence of its antioxidant and anti-inflammatory properties after digestion processes. To better stimulate the real effect of a possible functional food based on cinnamon extract, the obtained aqueous extract was subjected to simulate gastrointestinal digestion (i.e., INFOGEST protocol [13]), with the aim of evaluating the stability of the bioactive molecules and therefore the functionality of the digested extract at the cellular level. A semi quantitative chemical analysis on total extract and digested ones was performed to define quantitative changes on phyto-complexes. Moreover, cellular tests were performed to evaluate the antioxidant and anti-inflammatory activity of the digested samples, with the final aim of supporting the production processes of functional foods and to confirm the beneficial properties of cinnamon-based extracts.

## 2. Materials and Methods

### 2.1. Samples Preparation and In Vitro Digestion

*C. verum* L. dried barks (batch number: C1900010, date: 16 January 2019) obtained by EPO s.r.l., was pulverized with an electric laboratory grinder. The method used for the preparation of cinnamon extracts has been described by De Giani et al., 2022 [14]. Briefly, 1 g of powder was extracted with hot water by 1 h incubation in a rotavapor (Strike-300, Steroglass Italia Srl, San Martino in Campo, Italy). At the end of the process, a centrifugation was carried out at 600 rpm for 15 min to recover the supernatant. Then, 75% ethanol was added in a 1:1 ratio to precipitate polysaccharides. To improve and accelerate the precipitation process, the mixture was stored 1 h at 4 °C. Then, the extract was filtered under vacuum using Whatman filters n.1. Finally, the solvent was evaporated with rotavapor and the dry fraction was resuspended in 5–10 mL of water. The samples were freeze-dried and stored at −20 °C.

### 2.2. Gastrointestinal Digestion Process In Vitro

The gastrointestinal digestion simulation was performed using INFOGEST protocol according to Minekeus et al., 2014 [13]. The process was simulated in 3 phases: oral phase, gastric phase and intestinal phase.

Oral phase: 1.4 mL of the oral saline phase (SSF) previously prepared as in Minekeus et al., 2014, 200 µL of α-amylase 75 U/mL, 10 µL of CaCl_2_ and 390 µL of water were added to 2 mL of cinnamon water extract 10 mg/mL. The oral phase was set up at 37 °C for a duration of 2 min.

Gastric phase: 3 mL of the gastric saline phase as in Minekeus et al., 2014, 640 µL of swine pepsin 2000 U/mL, 2 µL of CaCl_2_ and 278 µL of water were added to the 4 mL of bolus obtained from the oral phase. Finally, the pH of the gastric mixture was acidified with 160 µL of HCl 2 M to reach pH 3. The gastric phase was set up at 37 °C for a duration of 2 h.

Intestinal phase: 8 mL of gastric chyme was mixed with 4.4 mL of the intestinal saline phase previously prepared as in Minekeus et al., 2014. The pH was then measured to ensure that it was neutral or slightly alkaline. At this point 14 mL of porcine bile, 2 mL of porcine pancreatin to obtain trypsin with activity of 100 U/mL, 16 µL of CaCl_2_ and 1.58 mL of water were added. The intestinal phase was set up at 37 °C for a duration of 2 h.

Upon completion of the intestinal phase, the digestion product was acidified to pH 2–3 to precipitate enzymes and to preserve the polyphenols as suggested by Pineda-Vadillo et al., 2016 [15]. Finally, to stop the enzymatic activity and to preserve the sample, protease inhibition was added.

### 2.3. Chemical Analysis of Polyphenols

The Folin–Ciocalteu assay was performed to determine the total polyphenol content of the samples as described in Pagliari et al., 2022 [16]. For the quantification of polyphenols, a calibration line was prepared using as a standard the gallic acid prepared at different operating concentrations, 100-75-50-25-12.5-6.25-0 mg/mL. Sample volume apart (i.e., 80 µL), the solvents and the respective volumes used in the assay were 400 µL of water, 40 µL of the Folin–Ciocalteu reagent and 480 µL of sodium carbonate at 10.75%.

The readings of the samples and the points of the line were taken at a wavelength of 760 nm after a half-hour wait, using a Cary 60 UV-Vis spectrophotometer (Agilent Technologies, Inc. Santa Clara, CA, USA). Results were expressed as mg [GAE]/g.

For the samples obtained from the in vitro digestion simulation, it was necessary to make some modifications to the Folin–Ciocalteu assay. The main reasons of this choice relied on the liquid physical state of samples and the need to reduce interference due to enzymes and salts used during the digestion process. For this purpose, the indications reported by Helal and Tagliazucchi, 2018, were followed [17]. Therefore, 150 μL of sodium carbonate 20% was used, while the volumes of water, sample and Folin–Ciocalteu reagent were 790 μL, 10 μL and 50 μL, respectively. The assay required an incubation of 2 h and was followed by a reading of the absorbance at a wavelength of 760 nm.

### 2.4. UHPLC-DAD-ESI-HRMS Profile of Cinnamon Extract and Digestion Product

The identification of target bioactive molecules was carried out using a Thermo Vanquish UHPLC system (Thermo Scientific, Rodano, Italy) coupled with a Thermo Orbitrap Exploris 120 mass spectrometer (Thermo Scientific, Rodano, Italy) and a Vanquish Diode Array Detector (Thermo Scientific, Rodano, Italy). The chromatographic separation was conducted on a Luna Omega Polar C18 (150 mm × 2.1 mm, 3 µm) (Phenomenex, Castelmaggiore, Italy).

The eluents, all MS grade from Fisher (Fisher scientific, Rodano, Italy), were (A) water and (B) acetonitrile/methanol (1:1), both containing 0.1% of formic acid. The gradient elution started from 2% of B, then raised at 40% in 40 min, at 50 min, the B concentration was 60% and at 55 min, 100% remaining constant for 10 min; finally, the mobile phase was brought back to the initial condition in 1 min. Moreover, 15 min of re-equilibration was necessary.

The flow rate was set at 300 µL/min, and the injection volume was 5 µL. The column was maintained at 30 °C. UV spectra were recorded between 210 and 600 nm, 280 and 340 nm wavelengths were employed, respectively, for the characterization of acids and flavonoids compounds. The UHPLC was coupled with the Orbitrap Exploris 120 through a HESI Optamax NG source. The source parameters were as follows: spray voltage 3500 V, ion transfer tube temperature 320 °C, vaporizer temperature 300 °C; sheath, auxiliary and sweep gas flow were 45, 10 and 0 arbitrary units, respectively. Spectra were recorded in full-mass mode within a range of 100–1500 *m*/*z* in positive and negative ionization, mild trapping condition was exploited. The Orbitrap resolution was set at 120,000 and the RF Lens (%) was 80. Data-dependent MS experiments were performed in stepped collision energy mode, the normalized collision energy was 20, 40 and 80%, HCD (higher energy collisional dissociation) fragmentation was achieved.

Phenolic compounds were characterized according to the corresponding spectral characteristics (UV and MS/MS spectra), accurate molecular mass, characteristic MS fragmentation pattern and libraries comparison in semi-automatic way through Compound Discoverer Software (2.1, Thermo Scientific, Rodano, Italy). Then, a semi-quantification was conducted in order to disentangle the phyto-complex composition change after digestion of the cinnamon extract. Briefly, all compounds were quantified, depending on their structural similarity, as trans-cinnamic acid (* in Appendix A) or catechin (** in Appendix A) at 280 nm by external calibration line. The quantification was performed in duplicate.

### 2.5. Caco-2 Cell Cultures

Cellular tests were performed on the Caco-2 (ATCC^®^ HTB-37™) human colorectal cancer cell line, which is a continuous line of heterogeneous human epithelial colorectal adenocarcinoma cells that, upon reaching confluence, express the characteristics of enterocytic differentiation: tight junctions, microvilli and a number of enzymes and transporters that are characteristic of enterocytes [18]. Caco-2 cells were grown in EMEM medium supplemented with 10% heat-inactivated fetal bovine serum (FBS), 2 mM L-glutamine, 0.1 mM non-essential amino acids, 100 U/mL penicillin, 100 µg/mL streptomycin. The Caco-2 cell line was maintained at 37 °C in a humidified 5% CO_2_ incubator. The ATCC cell line was validated by short tandem repeat profiles that are generated by simultaneous amplification of multiple short tandem repeat loci and amelogenin (for gender identification). All the reagents for cell cultures were supplied by Euroclone (Euroclone Spa, Milan, Italy). Caco-2 cells were seeded at a density of 1 × 10^4^ cells/well in 96-well microtiter plates, at 1 × 10^6^ cells/dish in 100 mm dish or at 90,000 cell/cm^2^ onto collagen coated Transwell^®^ polyester membrane inserts, based on the suitable conditions for oxidative stress and inflammation assays.

### 2.6. Intestinal Barrier In Vitro Model

The anti-inflammatory capacity experiments were performed in an in vitro intestinal barrier model since it represents the most suitable condition to study the gut maintenance and defense to inflammatory mediators and the modulation of intestinal epithelial permeability. Caco-2 cellswere seeded onto collagen coated Transwell^®^ polyester membrane inserts (Corning 3640) and maintained at 37 °C in a 5% CO_2_ atmosphere for 21 days, in order to reach the proper differentiation in a polarized epithelial cell monolayer. The basolateral and the apical sides of the insert represent the circulatory and luminal poles of the intestinal epithelium, respectively. Medium was replaced both in apical and basolateral chambers every 3 days, and Transendothelial Electrical Resistance (TEER) was measured every 7 days by means of EVOM EndOhm-12 chamber (World Precision Instruments, Sarasota, FL, USA), to check the formation of a reliable intestinal barrier.

TEER was evaluated as a parameter of barrier functionality after inflammation in cells pretreated or not with cinnamon digested extract. TEER measures were performed before and after the 24 h-inflammatory treatment for every insert, and the ΔTEER [Δ (Ω·cm^2^)] between the two measures was calculated and analyzed within the groups.

### 2.7. Intestinal Barrier In Vitro Model

Caco-2 cells were treated with two different concentrations in polyphenols content (23 µg/mL or 46 µg/mL) of digested cinnamon extract 24 h before the induction of pro-inflammatory or pro-oxidative stimuli.

In order to mimic the acute phase of intestinal inflammation, Caco-2 cells cultured onto Transwell^®^ inserts were exposed to a cocktail of inflammatory mediators (TNFα 10 ng/mL + IL-1β 5 ng/mL) applied at the basolateral side of the cells for 24 h, as described by Van de Walle et al. (2010) with minor modifications.

In order to induce oxidative stress, Caco-2 cells cultured in microplates or dishes were treated for 1 h with 1 mM H_2_O_2_ [19].

### 2.8. Cell Viability Assay

Cell viability was investigated using MTT-based in vitro toxicology assay kit (Sigma, St. Louis, MO, USA), according to manufacturer’s protocols. Briefly, MTT [3-(4,5-Dimethythiazol-2-yl)-2,5-diphenyltetrazolium bromide] assay is based on the reduction of tetrazolium salts to colored formazan compounds that occurs in metabolically active cells. Optical density was measured using a multi-detection microplate reader at a wavelength of 570 nm. Cell viability was expressed as a percentage against untreated cell lines used as controls. MTT was performed in cells subjected to pro-inflammatory or pro-oxidative stimuli in the respective culture systems.

### 2.9. Detection of Reactive Oxygen Species (ROS)

The generation of intracellular reactive oxygen species (ROS) was detected by the oxidation of 2′,7′-dichlorofluorescin diacetate (H_2_DCFDA) (Sigma Chemical Co., St. Louis, MO, USA), an indicator for both reactive oxygen species and nitric oxide (•NO). Caco-2 cell lines were seeded in 96-well black microtiter plates at a density of 1 × 10^4^ cells/well, cultured in complete medium and treated with 23 and 46 μg/mL of polyphenols for 24 h and 1 mM H_2_O_2_ for 1 h for positive control. To evaluate the protective role of the enzymatic digestion, after 24 h of treatment, cells were washed in PBS and incubated with 5 μM H_2_DCFDA in PBS for 30 min in the dark at 37 °C. After two washes in PBS, cells were treated with 1mM H_2_O_2_ for 1 h. The fluorescence (λ_em_ = 485 nm/λ_ex_ = 535 nm) was measured at 37 °C using a fluorescence microtiter plate reader (VICTOR X3) and analyzed by the PerkinElmer 2030 Manager software for Windows.

### 2.10. Enzyme Assay

To evaluate the effect of enzymatic digestion of cinnamon on enzyme activities, Caco-2 cell lines were seeded at 1 × 10^6^ cells/100 mm dish and treated for 24 h with 23 and 46 μg/mL of polyphenols and with 1 mM H_2_O_2_ for 1 h. The cells were rinsed with ice-cold PBS and lysed in 50 mM Tris-HCl pH 7.4, 150 mM NaCl, 5 mM EDTA, 10% Glycerol, 1% NP-40, containing protease inhibitors (1 μM leupeptin, 2 μg/mL aprotinin, 1 μg/mL pepstatin and 1 mM PMSF). Homogenates were obtained by passing 5 times through a blunt 20-gauge needle fitted to a syringe and then centrifuged at 15,000× *g* for 30 min at 4 °C. Supernatants were used to measure enzyme activities: superoxide dismutase (SOD) was assayed as previously described by Vance et al., 1972 [20]; catalase (CAT) was assayed according to Oldani et al., 2020 [21]; glutathione reductase (GR) according to Wang et al., 2001 [22]; glutathione peroxidase (GPx) as reported in Nakamura et al., 1974 [23]; glutathione-S-Transferase (GST) as previously described by Habig et al., 1974 [24]. Enzyme activities were expressed in international units and referred to protein concentration as determined by the Bradford method [25]. All of the assays were performed in triplicate at 25 °C in a Jasco V-550 Spectrophotometer and analyzed by the Spectra Manager (version 1.33.02) software for Windows.

### 2.11. Electrophoresis and Immunoblotting

Equal amounts (as protein) of homogenate were analyzed by SDS-PAGE electrophoresis on 10% polyacrylamide tris-glycine gels. Proteins were transferred to a nitrocellulose membrane (Amersham, GE Healthcare Europe GmbH, Milano, Italy) and revealed by immunoblotting with specific antibodies. Immunoblotting was performed using rabbit polyclonal anti-COX2 (1:1000) (#12282 Cell signaling technology, Danvers, MA, USA), rabbit polyclonal phospho p65-NFκB (Ser536) (1:1000) and rabbit polyclonal p65-NF-κB (1:1000) (#93H1 and #4764 Cell signaling technology), rabbit polyclonal anti-β-actin (1:1500) (A2066 Sigma-Aldrich, St.Louis, MO, USA) antibodies. Immunoreactive proteins were revealed by enhanced chemiluminescence (ECL) and semi-quantitatively estimated by LAS800 Image Station. Normalization in the same sample was carried out with respect to β-actin homogenate samples [26,27].

### 2.12. IL-8 ELISA Assay

In order to assess IL-8 secretion in the compartment mimicking blood circulation, extracellular media were collected from the basolateral chamber. IL-8 concentration was evaluated using commercially available ELISA assay (PeproTech EC, Ltd., London, UK) according to the manufacturer’s instructions. Samples concentration was quantified in pg/mL using the standard provided by the kit, after reading absorbance with FLUOstar Omega (BMG Labtech, Ortenberg, Germany) multi-detection microplate reader at a wavelength of 405 nm and a reference wavelength of 690 nm. Then, to facilitate comparison between groups and to understand the anti-inflammatory potential of cinnamon, data normalized by cell protein concentration were expressed in relative terms to the inflammatory treatment.

### 2.13. Statistical Analysis

All data have been expressed as mean ± SEM (standard error of the means). Regarding the total content of phenols in different extraction condition, the water extraction method was compared with the other hydroalcoholic condition using Dunnet’s test following one-way ANOVA calculation, a *p*-value < 0.05 was considered statistically significant, while for cell assay, the values were compared to the negative control (untreated cells) or positive control (oxidant or inflammatory stimulus) using the Dunn test following one-way ANOVA calculation. A *p*-value < 0.05 was considered to be statistically significant.

## 3. Results and Discussion

### 3.1. Samples Preparation and In Vitro Digestion

Several studies are available in the literature on the efficacy of aqueous and hydroalcoholic extracts of cinnamon, which can be correlated with the phenolic compound in the spice [7,8,10]. The first phase of the work involved the comparison of different extraction protocols to define the one most suitable for the study of antioxidant and anti-inflammatory activities in vitro after stimulation of the digestive process. The results reported in Table 1 show a higher % yield in hydroalcoholic extracts than in aqueous extracts. However, a comparison of total phenol content, using the Foli–-Ciocalteu spectrophotometric assay, shows that the extract obtained with the aqueous solvent is comparable to a 30% and 70% hydroalcoholic extract (*p*-value < 0.05) with significant values (*p*-value > 0.05) only at ethanol percentages of 50%.

Moreover all hydroalcoholic extracts are insoluble in the aqueous environment of the gastrointestinal tract, indicating that only the aqueous fraction is bioavailable for the cells [14]. Accordingly, for this study, it was decided to select the aqueous extract for the following steps due to its high total phenolic content, comparable to some hydroalcoholic extracts, and especially because it contains only the water-soluble fraction that is bioavailable to cells in the gastrointestinal tract.

### 3.2. UHPLC-DAD-HESI-HRMS Untargeted Analysis of Cinnamon Extracts

In order to quantify the active compounds in the aqueous extract of cinnamon, the complete characterization by UHPLC-DAD-HESI-HRMS was carried out, before and after the simulated digestion, in a completely non targeted way. The structural elucidation of the molecules was performed by using UV spectra, HRMS data (accurate molecular mass, isotopic pattern, MS/MS fragmentation spectra) and literature databases comparison. UHPLC-DAD (extracted at 280 nm) chromatograms between 0 and 40 min of cinnamon extract before (A) and after (B) digestion are reported in Figure 1.

The chromatogram (A) is extremely complex and rich in different chromatographic peaks. All the identified molecules belong either to the class of procyanidins or they are derivatives of phenolic compounds. Procyanidins are oligomers of catechin, epicatechin and their gallic acid esters most frequently linked, either 4–6 or 4–8 (B-type procyanidins) [28]. Structural variations to procyanidins oligomers may also occur with formation of a second interflavonoid bond by C-O oxidative coupling to form A-type, as shown in additional Appendix A.

During the first ten minutes, small and hydrophilic molecules such as: xylitol, raffinose and gluconic acid, quinic acid, malic acid, citric acid were eluted. All of them were identified on the basis of HRMS spectra and Compound Discoverer libraries. Furthermore, the chromatographic trace was characterized by two very intense peaks at 22.99 min (number 29a on chromatogram) and 37.35 (number 16 on chromatogram), attributed respectively to cinnamtannin B1 and *trans*-cinnamic acid, and by a broad peak (number 26) starting at 18 min and ending at 40 min. This wide-ranging peak was ascribable to the procyanidins with different degrees of polymerization (from 2 to 8). After digestion (Figure 1B), the complete disappearance of all catechin polymers was observed, while the *trans*-cinnamic acid and other phenolic compounds, such as 2-hydoxybenzoic acid, coumarin and hydroxybenzaldehyde also present in the extract, became the principal components in the digested sample.

Appendix A shows the identification, also supported by reference [17,29,30,31,32], of all compounds individuated in *Cinnamomum verum* extract and the digested sample, their retention time, exact and observed molecular mass and their MS/MS principal fragments.

### 3.3. Gastrointestinal Digestion Process

The second important result obtained in this study is that the digestion process produced a significant reduction in the amount of catechin and its polymers (tannins). It is known that catechins can be easily degraded by the digestive process. Concerning the transformations of phenols along the digestive tract, it is known that these compounds are quite resistant to the acidic environment, while they can be easily degraded to the alkaline environment of the intestine and by the enzymes present. Catechins, for example, can easily undergo autoxidative processes due to the alkaline pH of the intestinal tract and by the presence of bile salts and pancreatin, explaining the disappearance in the extract cinnamon digested [33,34]. Other phenolic compounds, on the other hand, such as cinnamic acid or simple phenols, are less prone to oxidative processes preserving themselves more. As shown in Appendix A, the content of *trans*-cinnamic acid, 2-hydroxybenzoic acid, coumarin and hydroxybenzaldehyde was very similar before and after the simulated digestion step. This reduction in total polyphenol content as a result of digestion is also confirmed by results obtained by spectrophotometric tests (Folin–Ciocalteu assay). Indeed, the total phenolic content observed before and after the digestion process showed that the aqueous cinnamon extract initially contained 1029.8 ± 24.1 µg of polyphenols, about 56 percent of the extract, while after digestion, in the gastrointestinal condition, the total phenol content decreased to 430.15 ± 12.5 µg, indicating a reduction of about 50 percent.

More in general, these data suggest that the occurrence of bioactive (i.e., the amount of water-soluble phytochemicals released at each of the stages of digestion from the complex food material that could be considered absorbable by the human body) may vary significantly due to digestion. Therefore, it becomes essential to evaluate the beneficial effects of food extracts following the digestive process, in order to assess their real potential for human health. For this purpose, some bioactivity in vitro experiments were conducted, choosing the human colorectal epithelial adenocarcinoma cell line Caco-2 because it represents an in vitro model of the human intestinal mucosa and these cells are widely used to study absorption of digested compounds [35].

### 3.4. Protective Effect of Digested Cinnamon Extracts Againstr Oxidative Stress

The effect of digested cinnamon extracts on Caco-2 cells was evaluated by determining cell viability using the MTT assay. As reported in Figure 2A, the digested cinnamon extract at 23 and 46 µg/mL polyphenol concentrations was not cytotoxic. The treatment with 1 mM H_2_O_2_ for 1 h reduced cell viability by 55% compared to the control; however, pre-treatment of Caco-2 cells with digested cinnamon extract at 23 and 46 µg/mL polyphenols for 24 h before adding H_2_O_2_ significantly increased cell viability compared to treatment with only H_2_O_2_. Therefore, the digested cinnamon extract shows a protective effects against H_2_O_2_- induced damage in Caco-2 cells (Figure 2A).

Total cytoplasmic ROS were evaluated using H_2_DCFDA, which is oxidized to DCF fluorescent probe. After treatment with a digested cinnamon extract at 23 and 46 µg/mL polyphenols, a significant decrease in ROS production was measured, confirming the protective effect (Figure 2B). A significant ROS increase was observed after treatment with H_2_O_2_ compared to the control; however, a pre-treatment with digested cinnamon extract at 23 and 46 µg/mL polyphenols for 24 h significantly decreased ROS generation as compared to H_2_O_2_-stressed cells, bringing the level of ROS close to the control values (Figure 2B). The inability of digested cinnamon extract to lower ROS levels to exactly the control values suggests that, in the presence of 1 mM H_2_O_2_, extract concentration is limiting even at 46 μg/mL. Rather than testing higher extracts concentrations, further work will be aimed at identifying key compound(s) endowed with the highest protective effect against oxidative stress. The antioxidant properties of cinnamon extract after digestion are attributable to the ability of phenolic constituents to quench reactive oxygen species. Cinnamic acid and coumarin, which increase after digestion, can scavenge superoxide anions and hydroxyl radicals as well as other free radicals [36].

### 3.5. Effect of Digested Cinnamon Extract on Antioxidant Enzyme Activity

In order to evaluate the effect of digested cinnamon extract on antioxidant enzyme activities in the Caco-2 cell line, it was assessed the activity of superoxide dismutase (SOD), catalyzing the dismutation of the highly reactive superoxide anion to H_2_O_2_, catalase (CAT), catalyzing the decomposition of hydrogen peroxide to water and oxygen, glutathione reductase (GR), a flavoprotein that catalyzes the reduction of glutathione disulfide (GSSG) to glutathione (GSH) with the participation of NADPH as an electron donor, glutathione peroxidase (GPx), a cytosolic enzyme that catalyzes the reduction of hydrogen peroxide to water and oxygen as well as the reduction of peroxide radicals to alcohols and oxygen, and glutathione S-transferase (GST), involved in detoxification mechanisms, via conjugation of reduced glutathione with numerous substrates.

Treatment of Caco-2 cells with 1 mM H_2_O_2_ for 1 h induced a significant increase in the enzyme activity of superoxide dismutase and glutathione S-transferase and a significant decrease in glutathione peroxidase activity, compared to the control. However, SOD and GST increase was reduced by a pre-treatment with a digested cinnamon extract at 23 and 46 µg/mL polyphenols for 24 h before adding H_2_O_2_, while GPx activity returned close to the control level (Figure 3). On the other hand, both CAT and GR activities did not change upon H_2_O_2_ treatment, suggesting that these enzymes are already effective in managing oxidative stress; the decrease in GPx activity suggests the most H_2_O_2_ is detoxified within the cytosol and that GPx activity is not sufficient to fully inactivate 1 mM H_2_O_2_; pretreatment with digested cinnamon extract at both concentrations showed a protective effect against H_2_O_2_ damage, by increasing GPx activity (Figure 3).

Mammalian cells have evolved antioxidant enzymes to protect against oxidative stress. A change in the level of antioxidant enzyme activity can be considered a sensitive biomarker of cellular response to oxidative stress [37,38]. Therefore, these data confirm the protective effect of digested cinnamon extract in Caco-2 cells. Phenolic compounds in digested cinnamon extract, such as trans-cinnamic acid and coumarin, may be responsible for antioxidant activity through various mechanisms: removing free radicals, binding metal ions, inhibiting enzymatic systems that produce free radical forms, increasing the concentration of biologically important endogenous antioxidants and inducing the expression of a variety of genes responsible for the synthesis of enzymes that reduce oxidative stress, such as superoxide-dismutase, catalase, glutathione peroxidase, glutathione reductase, glutathione-S-transferase, which represent a primary line of antioxidant protection [39].

### 3.6. Effect of Digested Cinnamon Extracts on Anti-Inflammatory Processes

Acute inflammation in the gut leads to disruption of tight junctions anchoring the epithelial cells which are part of the intestinal barrier. The loss of integrity and the increase of barrier permeability have, as important consequences, the alteration of microbiota and immune system homeostasis [8]. Since several polyphenols have beneficial effects on epithelial barrier dysfunctions [40], the potential anti-inflammatory power of digested cinnamon was evaluated in an intestinal barrier in vitro model (see Material and Methods) exposed to pro-inflammatory cytokines stimulus.

First of all, after polarization in Transwell inserts, a MTT assay was performed also under inflammatory stimulus in cells pre-treated or not with digested cinnamon. Increasing concentrations in polyphenols of digested cinnamon (23 µg/mL and 46 µg/mL) were administered to Caco-2 cells for 24 h in the apical compartment, then, the inflammatory stimulus was induced by adding the cytokine cocktail (TNFα/IL-1β) in the basolateral compartment for other 24 h. Afterward, an MTT assay was performed in Caco-2 on Transwell filters to evaluate the effects of single or combined treatments. None of the performed treatments induced cytotoxicity (Figure 4A).

In parallel, the intestinal barrier integrity was investigated by evaluation of transepithelial resistance (TEER). According to the increment of barrier paracellular permeability observed in inflammatory diseases of the gastro-intestinal tract, TEER was reduced in Caco-2 under inflammatory conditions with respect to untreated cells. Interestingly, pre-treatment with digested cinnamon extracts partially prevent the decrement of TEER in a dose-dependent manner (Figure 4B). Results are more evident when shown as intra-treatment ΔTEER and compared inter-groups (Figure 4C). Indeed, in the major concentration used (46 µg/mL), cinnamon can significantly prevent 25% of TEER loss during inflammatory treatment. No significant changes were observed after exposure to cinnamon alone. Taken together, these data suggest that this spice has a protective effect against the barrier disruption induced by inflammatory cytokines. The high amount of coumarin, cinnamic acid and derivatives, enriched after digestion, probably exerted the observed bioactivity. Indeed, rescue of barrier integrity was observed in presence of cinnamic acid when Ca-co-2 cells were treated with another inflammatory stimulus, such as LPS. In this case, cinnamic acid seems to act as a modulator of tight junction (TJ) protein expression, which is the major determinant of paracellular permeability [7]. On the other hand, it has been already demonstrated that several polyphenols and phenolic catabolites are able to partially reverse the TEER decrement induced by a pro-inflammatory cytokine cocktail in the Caco-2 monolayer [41]. Moreover, in a recent review, Bernardi and colleagues highlighted the role of polyphenols as intestinal permeability modulators both in vitro and in vivo studies [40].

### 3.7. Effect of Digested Cinnamon Extract on NF-κB Pathway and Its Targets in Cells Exposed to Pro-Inflammatory Cytokines

The intestinal barrier alterations are sustained and amplified by an excessive production of pro-inflammatory cytokines and chemokines secreted locally by immune cells and enterocytes [42]. Most of them are highly produced by the transcriptional activity of the nuclear factor-kappa B (NF-κB) in a sort of vicious cycle sustained by pro-inflammatory cytokines themselves (e.g., TNF-α). The NF-κB is a complex constituted by homo-heterodimers of p50, p52, p65 (Rel-A), c-Rel and Rel-B proteins, sequestered in the cytosol by its inhibitor IκBα. Under an inflammatory stimulus, the activation of the canonical pathway passes through the phosphorylation and the following proteasome degradation of the inhibitor, allowing the NF-κB p65 migration to the nucleus. Here, when activated by phosphorylation on Ser536, NF-κB p65 promotes its target gene expression. Accordingly, in polarized Caco-2 on the Transwell system, the exposure to TNF-α/IL-1β induced an increment of phosphorylated NF-κB about three times versus untreated cells (Figure 5). Interestingly, a reduction of 30% in p65-NF-κB phosphorylation, after inflammation, was detected in cells pre-treated with the major dose in polyphenols (46 µg/mL) of cinnamon digested extract, indicating an anti-inflammatory capacity of the phyto-compound. Indeed, many of the flavonoids revealed by LC-MS mass analysis, such as quercetin, kaempferol, rutin (flavonols) and apigenin (flavone), as well as coumarin, concurred in down-regulation of the pro-inflammatory NF-κB pathway [43] and in overproduction of chemokines [44].

In several cellular trials, the modulation of NF-κB is strictly associated to reduction of gene targets expression. Thus, to follow this pathway and considering that IL-8 secretion increased under pro-inflammatory stimulus [19,45,46], this marker was chosen to be assessed under the different experimental conditions. IL-8 was measured in the basolateral compartment, which mimics the blood circulatory district, and reported as concentration (pg/mL) in Table 2. As expected, inflammatory treatment induced the IL-8 secretion of about six times higher than untreated cells. Then, to facilitate the comparison among groups under inflammation and to understand the effect of cinnamon treatment, data were normalized by cell protein concentration and expressed in relative terms to inflammatory treatment (Figure 6A).

The pre-treatment with cinnamon digested extract at the higher concentration significantly reduced the amount of IL-8 released by about 25%. This result is in line with a previous study showing that cinnamon extract reduced IL-8 secretion in Caco2 cells under inflammatory stimulus [7]. Indeed, increasing evi-dence links the bioactivity of polyphenols, and in particular flavonoids, to the modulation of IL-8 expression and secretion [47].

These data are particularly interesting since IL-8 concentrations in plasma of patients with chronic gastrointestinal disease are also maintained high in the remissive phases of pathologies [48]. Therefore, a decrease of IL-8 production and secretion could be an important factor to prevent a strong acute shot.

In order to deepen the anti-inflammatory properties of digested cinnamon extract, in parallel, the ciclooxigenase-2 (COX-2) protein levels were analyzed. This enzyme, among the NF-κB target, represents one of the most studied key markers of inflammation, because it catalyzes the arachidonic acid conversion in the inflammatory lipid mediators prostaglandins (PGE2).

According to previous evidence indicating that COX-2 protein expression augmented in several diseases characterized by inflammatory processes and oxidative stress, cytokines administration in Caco-2 cells induced an increment of the protein levels by about 2-fold with respect to untreated cells (200%). Interestingly, the pre-treatment with the higher digested extract concentration in polyphenols (46 µg/mL) reduced by about 35% the COX-2 levels observed under a pro-inflammatory stimulus (Figure 6B).

As a NF-κB target, COX2 expression is strictly associated to the nuclear factor regula-tion. Indeed, several studies reported that polyphenols, including apigenin, quercetin, kaempferol and resveratrol, inhibit COX activity both at the transcriptional levels [49]. Thus, COX2 inhibition by polyphenols might result in a PGE2 pro-inflammatory mediators reduction.

Taken together, these results showed a moderate but significant anti-inflammatory effect of digested cinnamon extract, due to all the polyphenols contained in the extract that might synergistically act to reduce acute inflammation through the inhibition of NF-κB activity.

## 4. Conclusions

This study suggests that even plant extracts obtained through eco-sustainable processes, for example, with aqueous solvents, can maintain their beneficial properties for human health. The antioxidant and anti-inflammatory actions are also maintained following the digestion process. Even if simulated digestion approaches with static models have some limitations, such as the inability to evaluate dynamic processes such as secretion flow rate or gastric emptying, they are widely used, due in part to their simplicity, cost-effectiveness, reproducibility and reliability. Indeed, the results obtained allowed us to hypothesize that food enriched with cinnamon extract retains the bioactive properties of the spice also after digestion.

Based on these results, two fundamental points emerge, (i) it is not important to know only which are the bioactive molecules but the analysis of the phyto-complex as a whole is the element of value that must be considered in order to be able to assign a real beneficial property to a plant extract; (ii) cellular tests show that the bioactive compounds are able to interact with cellular processes and act by preventing stress factors involving inflammation and oxidation pathways, hypothesizing an active role of cinnamon to prevent and/or mitigate non-communicable diseases, such as tumors and a wide panel of degenerative diseases. These evaluations could be of interest for the producers of cinnamon-based products to encourage the manufacturing of fresh foods, that are not subjected to cooking and/or excessive chemical-physical stress. At the same time, it would be essential to test not only the extract but the whole food product by subjecting it to digestion, chemical analysis and bioactivity study. This would make it possible to claim assumptions based on solid scientific grounds and not only on hypothetical properties related to the single bioactive compounds present in a certain extract.

## Figures and Tables

**Figure 1 foods-12-00452-f001:**
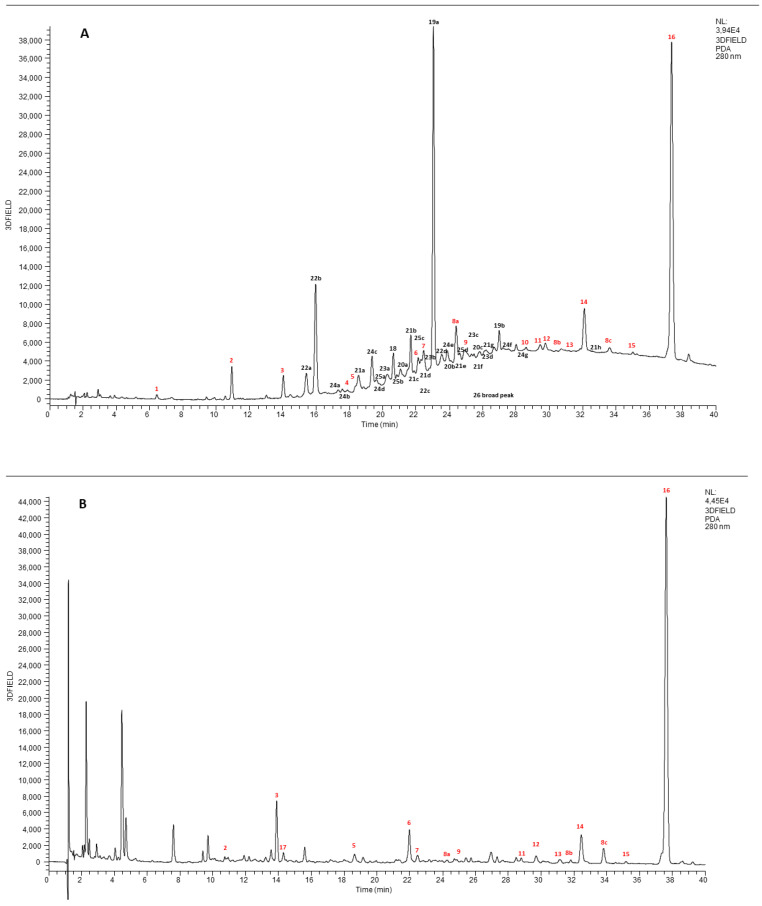
UHPLC-DAD chromatographic separation (280 nm) of cinnamon extract before (**A**) and after (**B**) digestion.

**Figure 2 foods-12-00452-f002:**
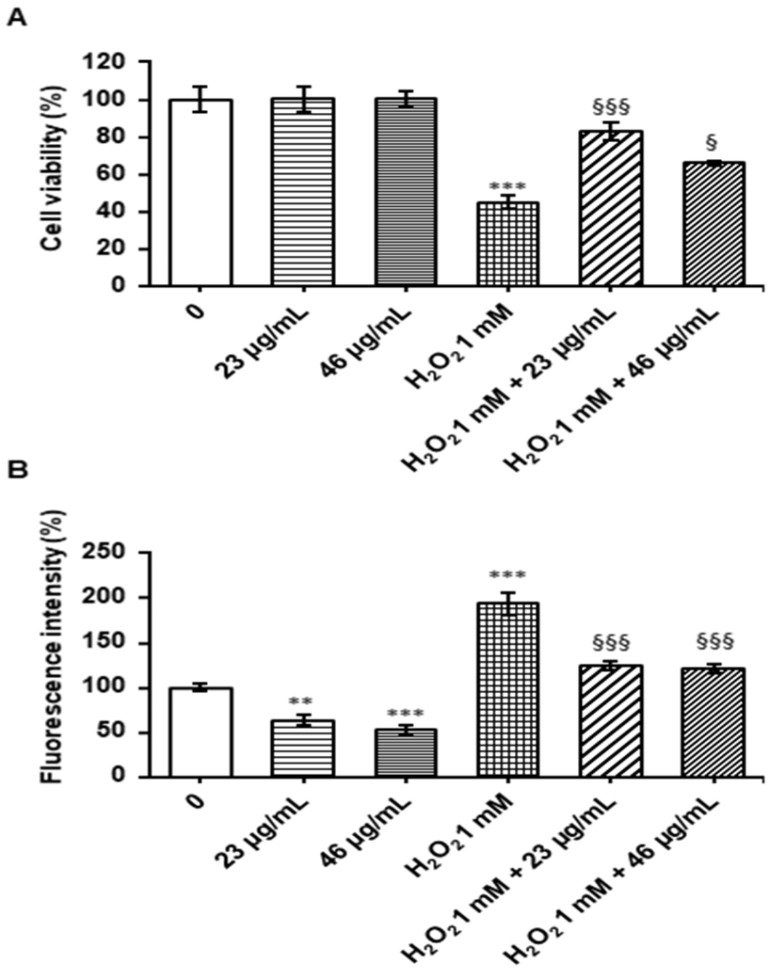
Viability assay and ROS evaluation. Caco-2 cells were treated with a digested cinnamon extract at 23 and 46 µg/mL polyphenols for 24 h, with 1 mM H_2_O_2_ for 1 h and with 1 mM H_2_O_2_ for 1 h after digested cinnamon extract pretreatment. In panel (**A**), the cell viability evaluated by MTT assay is reported; panel (**B**) shows the level of fluorescence after cells incubation with 5 μM H_2_DCFDA. Data are expressed as a percentage against the untreated cell line used as a control and are shown as mean ± SEM from three independent experiments. Statistical significance: ** *p* < 0.01, *** *p* < 0.001 respect to untreated cell line; § *p* < 0.05, §§§ *p* < 0.001 respect to H_2_O_2_-treated cell line.

**Figure 3 foods-12-00452-f003:**
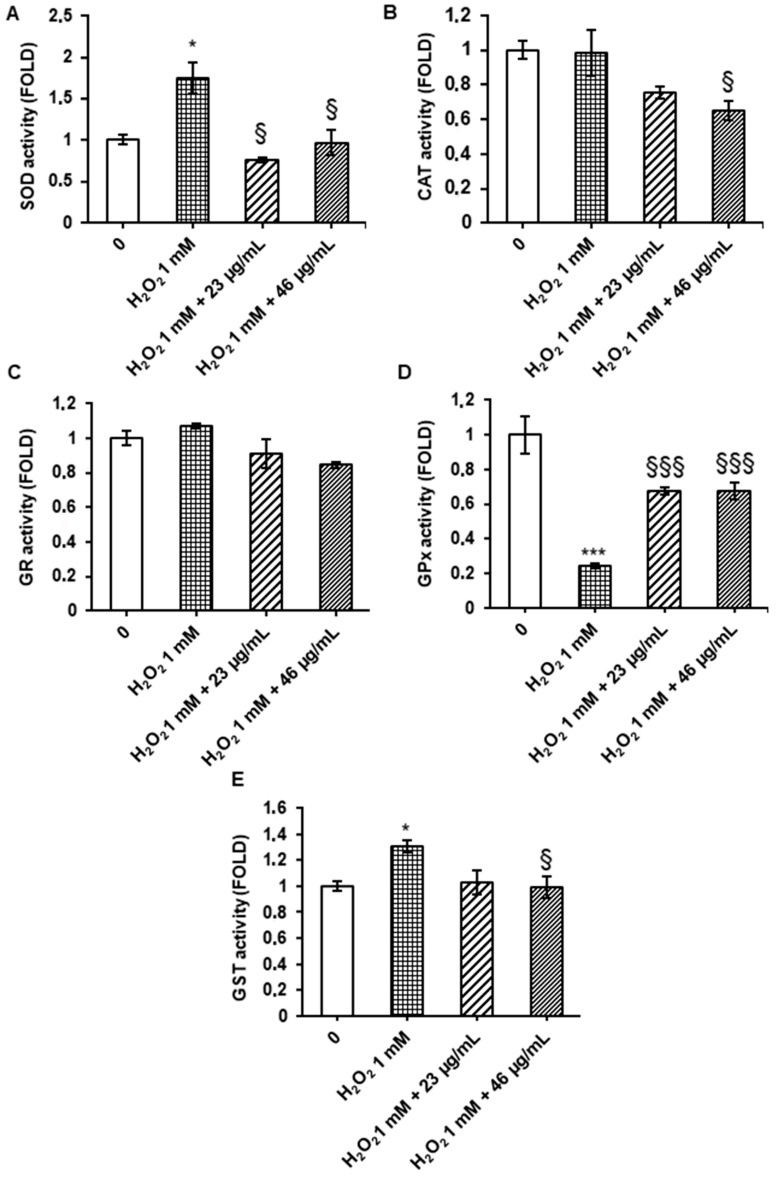
Enzyme activities. Caco-2 cells were treated with digested cinnamon extract at 23 and 46 µg/mL polyphenols for 24 h, with 1 mM H_2_O_2_ for 1 h and with 1 mM H_2_O_2_ for 1 h after digested cinnamon extract pretreatment. Results of SOD (**A**), CAT (**B**), GR (**C**), GPx (**D**) and GST (**E**) activities are expressed as folds with respect to untreated control and are shown as mean ± SEM from three independent experiments. Statistical significance: * *p* < 0.05, *** *p* < 0.001 respect to untreated cell line; § *p* < 0.05, §§§ *p* < 0.001 respect to H_2_O_2_-treated cell line.

**Figure 4 foods-12-00452-f004:**
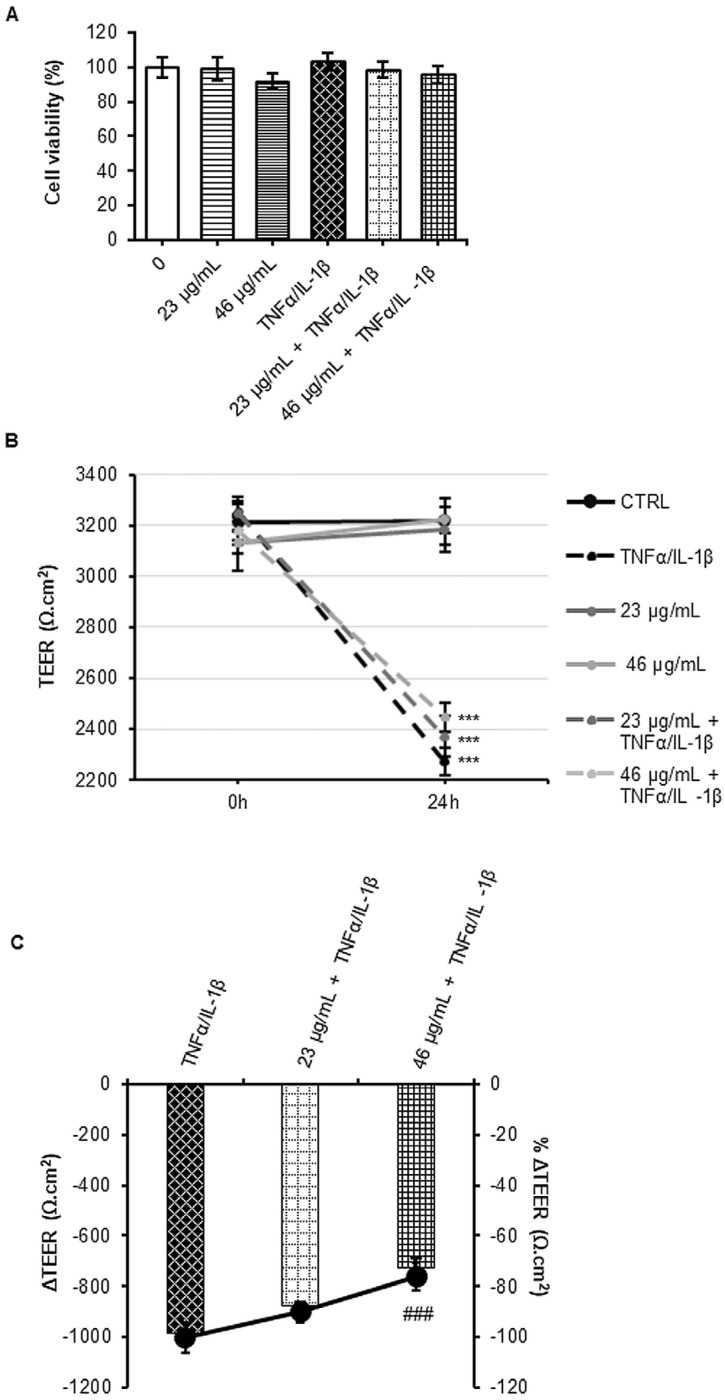
Cell viability assay and measurement of TEER under different experimental conditions. Caco-2 cells, cultured on Transwell inserts for 21 days were exposed to pro-inflammatory cytokines (TNFα 10 ng/mL + IL-1β 5 ng/ml) for 24 h after a 24 h pre-treatment with two different concentrations in polyphenols of digested cinnamon extract (23 and 46 µg/mL). Then, a MTT assay was performed, and results are reported in the histograms. Single conditions were also tested. Data are expressed as a percentage against the untreated cell line used as a control and are shown as mean ± SE from three independent experiments (**A**). TEER measures (Ώ·cm^2^) were performed before (0 h) and after (24 h) cytokines treatment (**B**). The ΔTEER of cells exposed to cytokines was calculated and reported in the histograms as number (left Y-axes) and percentage (right Y-axes). Data represent the mean ± SEM from at least three independent experiments (**C**). Statistical significance: *** *p* < 0.001 vs. untreated cells, ### *p* < 0.001 vs. TNFα/IL-1β.

**Figure 5 foods-12-00452-f005:**
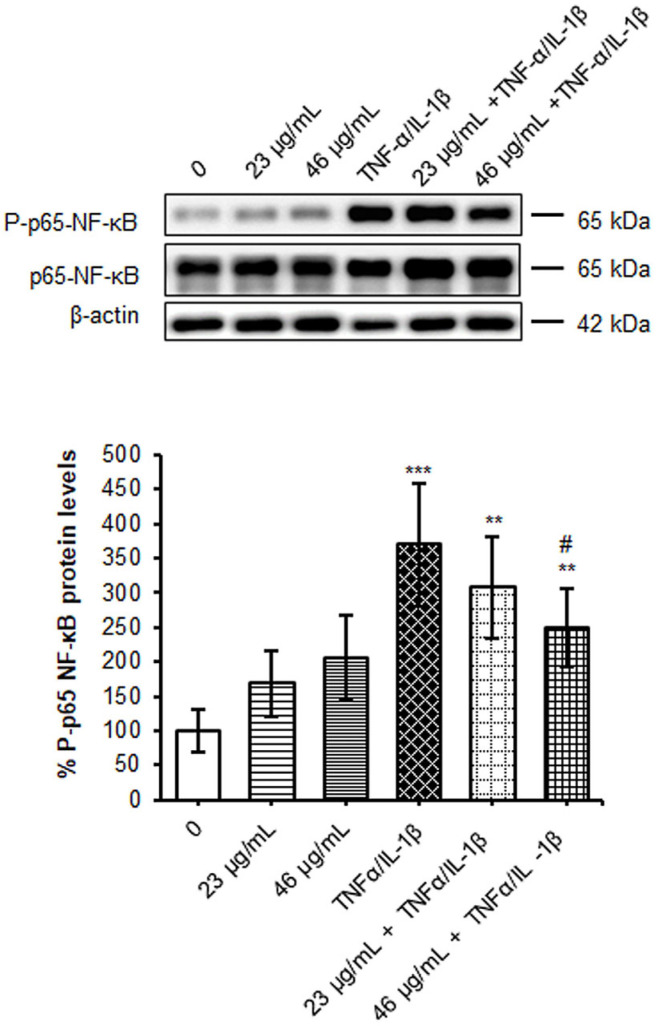
Evaluation of NF-κB phosphorylation and protein levels. Caco-2 cells, cultured on Transwell inserts for 21 days were exposed to pro-inflammatory cytokines (TNFα 10ng/mL + IL-1β 5 ng/mL) for 24 h after a 24 h pre-treatment with two different concentrations in polyphenols of digested cinnamon extract (23 and 46 µg/mL). Cell lysates were harvested, and samples analyzed for protein concentration by bicinchoninic acid (BCA) assay. Equal amounts of homogenate samples (as protein) were analyzed by SDS-PAGE electrophoresis and Western blotting. p-p65-NF-κB and NF-κB were detected with specific antibodies and revealed by enhanced chemiluminescence (ECL). Samples were normalized on β-actin immunoreactivity. Histograms, obtained from at least three distinct experiments, represent the percentage of protein levels with respect to untreated cells as mean ± S.E. Statistical significance: ** *p* < 0.01, *** *p* < 0.001 vs. untreated cells, # *p* < 0.05 vs. TNFα/IL-1β.

**Figure 6 foods-12-00452-f006:**
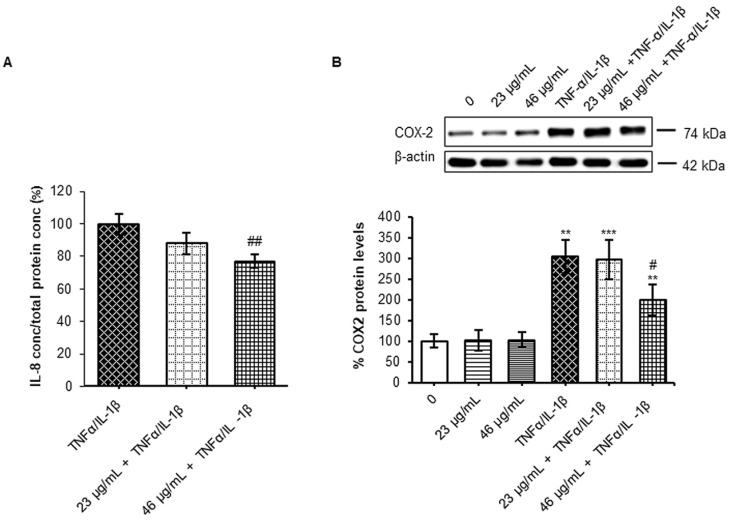
Evaluation of NFKB targets. Caco-2 cells, cultured on Transwell inserts for 21 days, were exposed to pro-inflammatory cytokines (TNFα 10 ng/mL + IL-1β 5 ng/mL) for 24 h after a 24 h pre-treatment with two different concentrations in polyphenols of digested cinnamon extract (23 and 46 µg/mL). Then, cell lysates and basolateral media were harvested. (**A**) IL-8 secretion was evaluated by ELISA assay. Data were normalized by cell total protein concentration. Results reported in histograms are expressed in relative terms to inflammatory treatment and are shown a mean ± SE from at least three independent experiments. Statistical significance: ## *p* < 0.01 vs. TNFα/IL-1β. (**B**) COX-2 protein levels were analyzed by SDS-PAGE electrophoresis and Western blotting. COX-2 was detected with a specific antibody and revealed by enhanced chemiluminescence (ECL). Samples were normalized on β-actin immunoreactivity. Histograms, obtained from at least three distinct experiments, represent the percentage of protein levels with respect to untreated cells as mean ± S.E. Statistical significance: ** *p* < 0.01 and *** *p* < 0.001 vs. CTRL and # *p* < 0.05 vs. TNFα/IL-1β.

**Table 1 foods-12-00452-t001:** Comparison between Folin–Ciocalteu assay among different *Cinnamomum* extract. The statistical analysis was carried out comparing water extraction conditions against other solvent ratios in the recovery of phenols compounds.

Sample	Solvent Condition	Yield (%)	Water Solubility	Folin–Ciocalteu Assay (mgGAE/gEXT)
**CIN_EtOH70_**	EtOH-H2O 70:30	10.57 ± 2.04	≤50%	517.39 ± 30.15
**CIN_EtOH50_**	EtOH-H2O 50:50	9.40 ± 1.64	≤50%	532.12 ± 20.61 **
**CIN_EtOH30_**	EtOH-H2O 30:70	7.06 ± 1.03	≤50%	480.18 ± 36.92
**CIN_H2O_**	H2O	5.44 ± 1.39	100%	482.87 ± 24.22

** *p*-value < 0.05.

**Table 2 foods-12-00452-t002:** Concentration (pg/mL) of IL-8 secreted in the basolateral media at the different conditions.

	0	23 µg/mL	46 µg/mL	TNFα/IL-1β	23 µg/mL + TNFα/IL-1β	46 µg/mL + TNFα/IL-1β
**pg/mL**	48.76 ± 4.74	39.96 ± 2.42	42.87 ± 7.7	322 ± 16.96	275.52 ± 20.49	266.60 ± 13.85
**Vs 0**				***	***	***

Results reported are expressed in relative terms to inflammatory treatment and are shown as mean ± SEM from at least three independent experiments. Statistical significance: *** *p* < 0.001 vs. untreated cells.

## Data Availability

All related data and methods are presented in this paper. Additional inquiries should be addressed to the corresponding author.

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
