# Peer review of "Antioxidant and Anti-Inflammatory Effect of Cinnamon (Cinnamomum verum J. Presl) Bark Extract after In Vitro Digestion Simulation"

_foods, 2023, doi:10.3390/foods12030452_

Round 1

Reviewer 1 Report

The study entitled "Antioxidant and anti-inflammatory effect of cinnamon (Cinnamomum verum J.Presl) bark extract after in vitro digestion simulation." shows the results of phytochemical analysis and antioxidant and anti-inflammatory evaluation of Cinnamomum verum plant extracts after simulations of digestive processes of the extract.

The manuscript contains interesting data aimed at proving the effectiveness of the active principles of a plant widely used in the food and medicine sector, after being subjected to chemical processes similar to the human digestive system. The text is properly written and presents important findings.

11)      On line 20, replace the name Cinnamomum with Cinnamomum verum J.Presl. It is important to specify the species that was studied by writing the full name.

22)      In line 22, the authors that cinnamic acid is phenolic. Cinnamic acid is not a phenolic compound. However, some of its derivatives are phenolic, such as caffeic acid, ferulic acid and p-coumaric acid. Please review the sentence.

33)      In line 23, authors mention pH as a physical agent. Please correct. This is a chemical property.

44)      On line 26, there is the phrase "cellular effect on vitro models". The experimental model is "in vitro". Please change the sentence accordingly.

55)      Please, I suggest inserting "natural product" in keywords.

66)      On line 38, it was written "Cinnamomum aromaticum

77)      or cassia". Authors should write full species names. For example, Cinnamomum aromaticum or Cinnamomum cassia.

88)      On line 45, it says "there are vanillic, caffeic, gallic, p-coumaric, ferulic". Authors should write the full names of the compounds. For example, there are vanillic acid, caffeic acid, gallic acid, p-coumaric acid. The "p" of the compound must be italicized.

99)      On line 78, if the product was purchased from EPO s.r.l., please enter the batch number, trademark and production date information.

110)   On line 300, write the term "acid" after the words "gluconic, quinic, malic".

111)   On line 312, write the scientific name instead of the common name of the cinnamon plant.

Author Response

On line 20, replace the name Cinnamomum with Cinnamomum verum J.Presl. It is important to specify the species that was studied by writing the full name.

Thank you for the advice, the name has been corrected in the text.

In line 22, the authors that cinnamic acid is phenolic. Cinnamic acid is not a phenolic compound. However, some of its derivatives are phenolic, such as caffeic acid, ferulic acid and p-coumaric acid. Please review the sentence.

Thank you, the sentence has been corrected.

In line 23, authors mention pH as a physical agent. Please correct. This is a chemical property.

Thank you, the sentence has been corrected.

On line 26, there is the phrase "cellular effect on vitro models". The experimental model is "in vitro". Please change the sentence accordingly.

Thank you, the sentence has been corrected.

Please, I suggest inserting "natural product" in keywords.

Thank you for the advice, “natural product” has been added in the keywords.

On line 38, it was written "Cinnamomum aromaticum or cassia". Authors should write full species names. For example, Cinnamomum aromaticum or Cinnamomum cassia.

Thank you, the sentence has been corrected.

On line 45, it says "there are vanillic, caffeic, gallic, p-coumaric, ferulic". Authors should write the full names of the compounds. For example, there are vanillic acid, caffeic acid, gallic acid, p-coumaric acid. The "p" of the compound must be italicized.

Thank you, the full names of the compound have been written in the text.

On line 78, if the product was purchased from EPO s.r.l., please enter the batch number, trademark and production date information.

Thank you, the batch number and the production date information have been added to the text.

On line 300, write the term "acid" after the words "gluconic, quinic, malic".

Thank you, the full names of the compound have been written in the text.

On line 312, write the scientific name instead of the common name of the cinnamon plant.

Thank you, the scientific name has been written instead the common name.

Reviewer 2 Report

Why did you not perform the simulated digestion using the Infogest protocol?

Why did you not evaluate the samples phenolic content by AUC using a chromatographic method? The method you used is imprecise and subject to various interferences

Why did you analyse the digest samples through an altered Folin and not through by running a blank condition using the digestion enzymes and salts? Once again this analysis should have been performed via AUC using a chromatographic method.

Section 2.10 refers to which assay? The transwell assay is only described after this section.

Why did you not use a Caco-2/HT29-MTX model in the membrane simulation model?

How was the inflammation performed in the membrane simulation model?

I can not concur with the reasoning for the usage of the water extract. It is natural that a hydroethanolic extract would have poorer solubility that a 100% water one. You should have removed the ethanolic partition of those extracts and use the best performing one. In comparison the water extract has the lowest yield and the lowest phenolic compound content.

What was the base value of the GI tract conditions folin?

Cytoprotective effect is not a correct naming. I would suggest cell metabolism impact and protection or something similar.

In figure 2A what is the cell viability value for the 46 ug/mL with H2O2 condition? It appears to be very borderline with the 30% threshold defined for cytotoxicity.

In figure 2B while you correctly interpret the ROS reduction observed in the presence of the extract you do not address the fact that while the extracts reduced H2O2 damage the value obtained was still above the one registered for the basal control. This should be addressed.

Where is the analysis of the CAT and GR activity data? This section lacks proper discussion.

Section 3.6 is confusing. First you say that you use the transwell model. But then you say you perform a MTT biocompatibility assay. How? MTT requires cells to be fixated. The cytokine cocktail is introduced in which compartment and for how long?

Revise figure 4B. The axis legend pre-cyto and post-cyto is not correct and acceptable. Additionally, the selected imagery of the figure makes it difficult to comprehend. Please fully revise the figure.

Section 3.6 lacks any significant form of discussion and is not acceptable to compare your results with LPS-induced damage when you use a complex interleukin cocktail.

Section 3.7 was performed in monolayer or in the trasnwell system. It is not clear.

This confusion becomes greater when you relate the NF-KB data with that of IL-8 in the basolateral compartment. The author has be very clear of how the assay was performed and of what is being analysed. Otherwise, it is impossible to ascertain the validity and logic of the data presented.

IL-8 results lack discussion with the existing literature regarding phenolics and their immunomodulatory potential.

COX-2 data lacks discussion versus the existing literature.

Author Response

Why did you not perform the simulated digestion using the Infogest protocol?

Thank you for your observation, the simulation digestion process was performed using INFOGEST protocol. The information was added in line 90 of the text.

Why did you not evaluate the samples phenolic content by AUC using a chromatographic method? The method you used is imprecise and subject to various interferences

The determination of the phenolic content was performed by Folin method due to its universality and versatility, but, also by a chromatographic method (HPLC-DAD extracted at 280 nm). The different phenolic components were identified by HPLC-HRMS and after, the single peak (instead of the total area under the chromatographic curve) of the DAD chromatogram recorded at 280nm, was quantified through a regression line using trans-cinnamic acid and catechina as standards. The data are presented in Supplementary Table S1.

Why did you analyse the digest samples through an altered Folin and not through by running a blank condition using the digestion enzymes and salts? Once again this analysis should have been performed via AUC using a chromatographic method.

Thank you for your observation, the Folin method is frequently used to preliminarily evaluate the total content of polyphenols in extracts. Other papers (i.e. ref. 17) using this Folin essay on post-digestion samples are available. The Folin data in this work was performed primarily for the initial screening of the matrices and as a support to the more complete mass analysis which evaluated the amount for each metabolite in both the extract and the digestion product. However, as the cinnamomum extract, the digest sample was quantified by HPLC-DAD (the results are present in Supplementary Table S1).

Section 2.10 refers to which assay? The transwell assay is only described after this section.

We agree with the reviewer and we apologize for the text misassembling: the section 2.11 should have been after section 2.5. Now it has been moved and renamed as 2.6 section. The following sections were consequently re-named.

Why did you not use a Caco-2/HT29-MTX model in the membrane simulation model?

We agree with the reviewer about the use of a Caco-2/HT29-MTX model to mimic the gut barrier. Nevertheless, in this paper we chose to use the Caco-2 cells transwell model, a diffused in vitro model for these kind of studies (Iglesias et al., 2022-DOI: 10.1002/mnfr.202101033; Valdez et al., 2020-DOI: 10.1016/j.abb.2020.108409), in order to evaluate the effects of cinnamon digested extract directly in the enterocytes and maintain the same cellular contest performed for oxidative stress.

How was the inflammation performed in the membrane simulation model?

We thank the reviewer for his question. Actually we described the inflammatory treatment protocol in the section 2.5 lines 178-183. Nevertheless, to improve the readability, we decided to divide section 2.5 in two different section: 2.5 Caco-2 cell culture and 2.7 Cell treatments in which we described the inflammation protocol lines 198-201. In the middle, we inserted the section about transwell system.

I can not concur with the reasoning for the usage of the water extract. It is natural that a hydroethanolic extract would have poorer solubility that a 100% water one. You should have removed the ethanolic partition of those extracts and use the best performing one. In comparison the water extract has the lowest yield and the lowest phenolic compound content.

Thanks for your observation, the choice of an aqueous extract was made by evaluating several aspects. Numerous studies are already available in the literature which characterizes cinnamon bark extracted with different solvents, the phenolic fraction which contributes to the activity of the spice is already widely known. The present work aims instead to complete the information available on the spice by evaluating the efficacy of extracts already known as active following the digestive process and considering that cinnamon is often used as bark, in the aqueous environment of the gastrointestinal tract and cells only the water-soluble fraction is made bioavailable for enzymes and absorption. Furthermore, the comparison with hydroalcoholic extracts, often used for a good recovery of phenolic compounds, showed that the aqueous extract shows a high phenolic content comparable with hydroalcoholic extracts at 30% or 70%. Finally, the hydroalcoholic extracts showed, in all the conditions tested, poor solubility in water with the impossibility of administering the digested extract to the cells to evaluate its activities. Therefore, it was preferred to select a phenolic recovery procedure in water, already used in the literature and with proven therapeutic activities [14,31], to obtain a completely water-soluble and bioavailable extract for the intestinal cells under study.

What was the base value of the GI tract conditions folin?

Sorry for the unclearly, the value of the GI tract condition measuring with folin was 430.15 ug of polyphenols for the sample. The sentences in the text was modified to clarify the comprehension.

Cytoprotective effect is not a correct naming. I would suggest cell metabolism impact and protection or something similar.

We agree with the Reviewer about the term cytoprotective not being appropriate; however, section 3.4 does not describe a metabolic impact, but rather a protective effect against ROS. We therefore substituted “cytoprotective” with “protective effect against oxidative stress”.

In figure 2A what is the cell viability value for the 46 ug/mL with H2O2 condition? It appears to be very borderline with the 30% threshold defined for cytotoxicity.

The Reviewer is right, as the cell viability value for the 46 ug/mL with H2O2 condition is 65.93%, a value which is below the 30% threshold defined for cytotoxicity; however, although there is a weak cytotoxic effect, the increase in cell viability compared to the treatment with H2O2 alone is statistically significant. However, pretreatment with 23 ug/mL cinnamon extract was found more effective in restoring cell viability. The reason why is not entirely clear, but it is difficult to find out at this stage, since the cinnamon extract is a mixture of different components. Thus, a higher extract concentration, although not toxic to cells, may hamper the protective effect of a particular compound. Further studies will be necessary to identify the most effective component.

In figure 2B while you correctly interpret the ROS reduction observed in the presence of the extract you do not address the fact that while the extracts reduced H2O2 damage the value obtained was still above the one registered for the basal control. This should be addressed.

The Reviewer is right; we apologize for not discussing this point and we have added a paragraph in the revised manuscript. In fact, digested cinnamon extract at either concentration was not able to lower H2O2 concentration to control cells level. This suggests that, in the presence of 1 mM H2O2, extract concentration is limiting even at 46 ug/mL; further work will be aimed at identifying key compound(s) endowed with the highest protective effect against oxidative stress.

Where is the analysis of the CAT and GR activity data? This section lacks proper discussion.

The Reviewer is right; we apologize for not discussing this point and we have added a paragraph in the revised manuscript. Both CAT and GR activities do not change upon H2O2 treatment, suggesting that these enzymes are already effective in managing oxidative stress; the decrease in GPx activity suggests the most H2O2 is detoxified within the cytosol and that GPx activity is not sufficient to inactivate 1mM H2O2; pretreatment with digested cinnamon extract shows a protective effect against H2O2 damage, by increasing GPx activity.

Section 3.6 is confusing. First you say that you use the transwell model. But then you say you perform a MTT biocompatibility assay. How? MTT requires cells to be fixated. The cytokine cocktail is introduced in which compartment and for how long?

We thank the reviewer for the comment. Thus we changed the sentences in order to improve the readability and make the information more clear. Lines 476-482.

Revise figure 4B. The axis legend pre-cyto and post-cyto is not correct and acceptable. Additionally, the selected imagery of the figure makes it difficult to comprehend. Please fully revise the figure.

We thank the reviewer for his suggestion. The figure 4B was revised and the new version was inserted in the manuscript. Also figure legend was modified.

Section 3.6 lacks any significant form of discussion and is not acceptable to compare your results with LPS-induced damage when you use a complex interleukin cocktail.

Discussion was modified and amplified in order to accomplish the reviewer’s suggestion.

Section 3.7 was performed in monolayer or in the trasnwell system. It is not clear. This confusion becomes greater when you relate the NF-KB data with that of IL-8 in the basolateral compartment. The author has be very clear of how the assay was performed and of what is being analysed. Otherwise, it is impossible to ascertain the validity and logic of the data presented.

According to the reviewer we better clarify the experimental condition and analysis.

IL-8 results lack discussion with the existing literature regarding phenolics and their immunomodulatory potential.

Thank you for disclosing this point. Discussion was implemented as request.

COX-2 data lacks discussion versus the existing literature.

Thank you for disclosing this point. Discussion was implemented as request.

Reviewer 3 Report

This manuscript discussed the phenolics of cinnamon bark and their antioxidant and anti-inflammatory properties. However, authors should provide a potential explanation how and which particular compounds might be involved to show these properties. My other comments are:

1.      Title, Cinnamomum verum should be italic

2.      Line 45, p should be italic in p-coumaric

3.      Table 1, statistical analysis should be included

4.      Line 118, “Results were expressed as mg [GAE]/g”, then it was expressed as ugGAE/mg in Table 1?

5.      Lines 304 307, and 325, trans should be italic

Author Response

This manuscript discussed the phenolics of cinnamon bark and their antioxidant and anti-inflammatory properties. However, authors should provide a potential explanation how and which particular compounds might be involved to show these properties.

We thank the Reviewer for the suggestion and accordingly we have added a paragraph in the revised manuscript.

My other comments are:

Title, Cinnamomum verum should be italic

Thank you, the sentence has been corrected.

Line 45, p should be italic in p-coumaric

Thank you, the sentence has been corrected.

Table 1, statistical analysis should be included

Thank you for your suggestion, the statistical analysis was been added to Table 1.

Line 118, “Results were expressed as mg [GAE]/g”, then it was expressed as ugGAE/mg in Table 1?

Thank you, the results in Table 1 have been converted to mg[GAE]/g to unify the text.

Lines 304 307, and 325, trans should be italic

Thank you, the sentence has been corrected.

Round 2

Reviewer 3 Report

I think the authors have now revised the manuscript accordingly, and it is ready for the next step.